# Implementation and Validation of an OpenMM Plugin for the Deep Potential Representation of Potential Energy

**DOI:** 10.3390/ijms25031448

**Published:** 2024-01-24

**Authors:** Ye Ding, Jing Huang

**Affiliations:** 1College of Life Sciences, Zhejiang University, Hangzhou 310027, China; dingye@westlake.edu.cn; 2School of Life Sciences, Westlake University, Hangzhou 310024, China; 3Westlake AI Therapeutics Lab, Westlake Laboratory of Life Sciences and Biomedicine, Hangzhou 310024, China

**Keywords:** molecular dynamics, OpenMM, deep potential, machine learning potential, force field

## Abstract

Machine learning potentials, particularly the deep potential (DP) model, have revolutionized molecular dynamics (MD) simulations, striking a balance between accuracy and computational efficiency. To facilitate the DP model’s integration with the popular MD engine OpenMM, we have developed a versatile OpenMM plugin. This plugin supports a range of applications, from conventional MD simulations to alchemical free energy calculations and hybrid DP/MM simulations. Our extensive validation tests encompassed energy conservation in microcanonical ensemble simulations, fidelity in canonical ensemble generation, and the evaluation of the structural, transport, and thermodynamic properties of bulk water. The introduction of this plugin is expected to significantly expand the application scope of DP models within the MD simulation community, representing a major advancement in the field.

## 1. Introduction

Molecular dynamics (MD) simulation is a technique that is widely used to study many-particle systems in chemistry, biology, and materials science [1,2,3]. To fully unleash the explanatory and predictive power of MD simulations, the underlying potential energy models need to be both accurate and computationally efficient. The potential energy can be calculated on-the-fly by electronic structure methods, with the simulation system propagated with either the Ehrenfest or Car–Parrinello dynamics [4]. Due to the expensive computational cost associated with ab initio calculations, this is often only possible for material systems that are homogenous and can be represented by small numbers of atoms with proper periodic boundary conditions. For more heterogeneous chemical and biological systems, classical force fields (FFs) are often used, which include the harmonic bond, angle, and dihedral potentials to represent short-range bonded interactions and the Coulomb and Lennard–Jones potentials to model long-range non-bonded interactions. These systems also require significantly longer simulation times to explore their complicated conformational spaces. Decades of research have been invested to make these simulations feasible in terms of enhanced sampling techniques [5], highly optimized simulation software [6,7,8,9], and adaptation for heterogeneous computer hardware such as GPU, FPGA, and ASICs [10,11,12,13].

An emerging solution to circumvent the trade-off between accuracy and efficiency would be machine learning potentials (MLPs) [14,15], either kernel-based [16,17,18,19,20,21] or neural network-based [22,23,24,25,26,27,28,29,30,31,32,33,34,35,36,37]. A key factor of machine learning potentials is to design the input features that represent the symmetries of simulation systems as much as possible [17]. The first influential neural network-based potential model was proposed by Behler and Parrinello in 2007 [25]. They suggested that the total potential energy of a simulation system $U_{\mathrm{sys}}$ can be constructed as the summation of the atomic energy $U_{i}$ of each particle, see Equation (Equation 1). $U_{i}$ is determined by the particle type and its surrounding environment with individual neural network $NN_{i}$ fed by an environment feature vector. This vector is constructed from the particle’s cartesian coordinates {R} and fulfills the rotational symmetry of molecular systems.
(1)$$U_{\mathrm{sys}}=\sum_{i=1}^{N}U_{i}=\sum_{i=1}^{N}NN_{i}\left(\left\{R\right\}\right).$$

A variety of neural network potentials (NNPs) have been proposed since then, with different network architectures and input features. Smith et al. developed the ANI-1 series of NNP models [38,39] by introducing more sophisticated three-body atomic environment features to better account for heterogeneous molecular environments. The message passing neural network (MPNN) [40] fetched the atomic environment information by recursively iterating the neighbor atoms along the topological links with a cost of reduced computational efficiency. This approach has been employed in various NNP models such as the deep tensor neural network (DTNN) [30] and SchNet [31].

Another approach to improve the accuracy of NNPs involves using embedding networks that learn the weighting information of surrounding particles and automatically construct atomic environment vectors. Zhang and co-workers proposed the deep potential (DP) model that constructs the potential using two different types of neural networks [28]. First, an embedding network takes the reciprocal distances between the central atom and its surrounding particles as input, and outputs an embedding vector that captures the weighting information of surrounding particles. An embedding matrix is then assembled by concatenating the embedding vectors of neighbor particles ordered by their reciprocal distances and further multiplied by the distance matrix and its transpose. The embedding matrix is thus invariant to the translation, rotation, and permutation of atoms, and serves as the input of a second fitting network to output the atomic energy $U_{i}$. The introduction of the embedding network in the DP model eliminates the need for empirically designed atomic-centered symmetry functions and improves generalization performance. We note that other types of effective embedding and fitting mechanisms have been proposed [41].

The DP model strikes a balance between accuracy and computational cost and has been widely used in the simulations of complex systems, in particular homogenous systems [42,43,44]. The lower computational cost of the DP model enables the capability of ∼ns/day simulation performance for a copper system consisting of more than 100 million atoms in a Summit supercomputer, with an ab initio-level accuracy [45]. Furthermore, with its superior generalization ability, a unique DP model can be utilized to explore the phase diagram of water over a vast range of temperatures and pressure with high-level accuracy [44]. Not only does the DP model provide an accurate description of the energy landscape for a wide range of chemical space, but it also enables simulations of systems that are otherwise inaccessible through ab initio calculations [43]. However, its applicability in heterogeneous systems, particularly in biomolecular systems, is limited due to challenges associated with the proper inclusion of long-range interactions and efficient sampling of conformation spaces.

The community of the DP model continues to evolve, with ongoing developments aimed at enhancing its capabilities. For instance, one notable advancement involves the incorporation of explicit long-range electrostatic interactions through the use of Gaussian charges, which are modeled by an additional deep neural network [46,47]. This would make it more suitable for modeling heterogeneous systems such as biological macromolecules. In a recent study, Pan et al. utilized the deep potential method to systematically improve the accuracy of the QM/MM free energy calculations of enzyme reactions by replacing the QM part with a DP model [48]. The DP method was originally implemented in a high-performance tool DeePMD-kit [49] for both model training and model inference, with an interface to LAMMPS https://www.lammps.org/#gsc.tab=0 (accessed on 1 December 2023) [50] for running MD simulations. With more application scenarios and functionalities developed by the DP model, an efficient and flexible integration of the DP model into more MD engines would be desirable for the MD simulation community.

MLPs were usually implemented in individually developed packages, most of which have limited functionalities. Efforts to implement different ML potentials into one software package have been attempted, for example in the MLatom package [51,52]. The separate implementation of ML potentials would help validate and benchmark these potential models. It would also be beneficial to implement ML potentials into existing MD engines such as CHARMM [7], GROMACS [9], Amber [6], and OpenMM [10]. This would allow the ML potentials to be combined with decades of efforts to realize accurate and advanced integration, free energy, and enhanced sampling techniques. The implementation of ML potentials in these traditional MD codes would also greatly promote them to be more broadly used and allow direct integration with existing empirical FF models.

OpenMM is a high-performance MD engine that supports multiple hardware platforms, in particular GPUs [10,53,54,55]. It was designed to be hierarchical and extendable, and it supports almost all modern GPUs. OpenMM implements a user-friendly interface with Python and key MD algorithms with C++ and CUDA for performance. It supports a wide range of empirical FF models including the polarizable AMOEBA [56] and Drude FFs [57], the atom-typing-free open force field models [58] as well as some machine learning potentials via its TensorFlow plugin [59] and Torch plugin [60]. Although the DP model was implemented in the DeePMD-kit package that is based on TensorFlow, the DP model cannot be running with the current TensorFlow plugin in OpenMM due to the wrapped C/C++ interface for inference in the DeePMD-kit package. Similar works are still ongoing for the efficient integration of OpenMM and NNPs [61].

In this study, we introduce and validate an efficient OpenMM plugin that provides support for DP models. This plugin incorporates several advantageous features, which allow for performing free energy perturbation (FEP) calculations and running multiscale DP/MM simulations with fixed or adaptive DP regions [62]. We validated the plugin using a comprehensive range of kinetic and thermodynamic characteristics using MD simulations with the DP model. Validation tests include the evaluation of energy conservation in the NVE ensemble, the structural and dynamical properties of bulk waters, as well as the fidelity of NVT ensemble generation. Moreover, we propose a protocol to carry out alchemical free energy calculations using the DP model through our plugin, exemplified by calculating the hydration free energy of water molecules. To substantiate the efficacy of DP/MM simulations, we performed two distinct simulation experiments, one featuring a fixed DP region for water molecules and another employing an adaptive DP region for a zinc-containing protein system. Collectively, our findings attest to the effectiveness and adaptability of our newly developed OpenMM plugin for simulations with the machine learning DP models.

## 2. Results

### 2.1. Design and Implementation of the OpenMM Deepmd Plugin

#### 2.1.1. General Architecture

The OpenMM Deepmd plugin serves as a flexible and efficient interface connecting OpenMM with the DP model. Its initialization involves utilizing the DeepPotentialModel class, enabling the loading of trained DP models. During each MD step, the plugin retrieves particle coordinates from the OpenMM context and passes them to the DP model for energy and force inferences (see Figure 1). To broaden the plugin’s applicability and enhance user flexibility, the selected particles can encompass either the entire system or a specific subset. The plugin also supports the selection of particles based on their proximity to designated central particles, and it updates the OpenMM context with the newly calculated energy and forces. This integration not only allows the usage of the DP model in conventional MD simulations but also facilitates its utilization in hybrid MD simulations with other FF models. The current implementation of the Deepmd plugin ensures that it can be employed on both CPU and GPU platforms. While the GPU version incurs additional communication overhead between the CPU and GPU at each step, our profiling data presented in Section 2.2.6 reveal that this overhead represents only a minor fraction of the total per-step computational cost.

#### 2.1.2. Python Class DeepPotentialModel

To facilitate the utilization of the deep potential model in OpenMM with this plugin, we have carefully designed a Python wrapped class named DeepPotentialModel. This class serves as a wrapper around the raw Python class DeepmdForce, thereby providing a more user-friendly interface for employing the deep potential model within OpenMM. The DeepPotentialModel class presents users with five distinct methods; the first two are essential for all application scenarios, while the remaining three are optional. The initialization of DeepPotentialModel only requires the path to the DP model file. Users have the additional option to specify a scale factor λ, which scales the forces and energies exerted by the DP model within the simulation. By default, λ is set to 1.$$\begin{array}{c} \mathrm{dp}\_\mathrm{model}=\mathrm{DeepPotentialModel}(\mathrm{model}\_\mathrm{file},\mathrm{Lambda}=1.0). \end{array}$$

While the variable units used in the training and inference of DP models are generally flexible, OpenMM imposes more strict unit requirements: nanometers (nm) for distance, kilojoules per mole (kJ/mol) for energy, and kilojoules per mole-nanometer (kJ/(mol·nm)) for force. The disparity in units requires users to explicitly set conversion factors. Specifically, three coefficients must be specified for the conversion of coordinate units, force units, and energy units, respectively. These coefficients do not have predefined values since they are determined by the training data used to construct the DP model. This also underscores the importance for users to ensure that the units of their training data are compatible with the OpenMM context before initiating simulations. The coord_coefficient acts as a multiplicative factor applied to particle coordinates before feeding them into the DP model. Similarly, the force_coefficient and energy_coefficient will be respectively applied to the forces and energy values calculated by the DP model before passing them to the OpenMM context. $$\begin{array}{c} \mathrm{dp}\_\mathrm{model}.\mathrm{setUnitTransformCoefficients}(\;\;\mathrm{coord}\_\mathrm{coefficient},\mathrm{force}\_\mathrm{coefficient},\mathrm{energy}\_\mathrm{coefficient})\;. \end{array}$$

The createSystem method provides a straightforward approach to preparing an OpenMM simulation context with this plugin. In particular, the method generates an OpenMM System object using the simulated system’s topology as its input. This object can then be used to construct an OpenMM Simulation object for subsequent simulations. It is important to note that the System object generated through this method is indistinguishable from those created with the native classical force fields present in OpenMM. $$\begin{array}{c} \mathrm{dp}\_\mathrm{system}=\mathrm{dp}\_\mathrm{model}.\mathrm{createSystem}(\mathrm{topology}). \end{array}$$

These three methods are already sufficient to construct a conventional simulation context employing the DP model within OpenMM, although the utility of the third method is not always needed. Recent studies, especially in the realm of biomolecular systems, show that most applications of MLPs in molecular dynamic simulations are based on hybrid MLP/MM frameworks [48,61,62,63,64,65] In such schemes, the particles described by the MLP models constitute a subset of the entire system. For implementing the hybrid MLP/MM scheme in OpenMM, the DeepPotentialModel class offers two optional methods that allow users to specify which particles will be input to the DP model and generate the Force object designated for integration with classical force fields. The major distinction between these two methods lies in how they handle the selection of input particles for the DP model: one method exclusively uses the specified particles as input, while the other incorporates both the specified particles and their adjacent particles. To explicitly define the input particles for the DP model, both the topology of the entire system and the list of these particle indices, denoted as dp_particles, are required. $$\begin{array}{c} \mathrm{dp}\_\mathrm{force}=\mathrm{dp}\_\mathrm{model}.\mathrm{addParticlesToDPRegion}(\mathrm{dp}\_\mathrm{particles},\mathrm{topology}). \end{array}$$

To specify the input particles for the DP model along with their neighbors, the topology of the entire system, the center particle indices center_particles, and the cutoff distance radius are needed. By default, the radius is set to 0.35 nanometers. Furthermore, for a reasonable selection of neighboring particles in biomolecular systems, the particles selected based on distance are, by default, extended to their entire residues through extend_residues=True. $$\begin{array}{c} \mathrm{dp}\_\mathrm{force}=\mathrm{dp}\_\mathrm{model}.\mathrm{addCenterParticlesToAdaptiveDPRegion}(\;\mathrm{center}\_\mathrm{particles},\mathrm{topology},\mathrm{radius}=0.35,\mathrm{extend}\_\mathrm{residues}=\mathrm{True}). \end{array}$$

This combination of methods significantly expands the range of practical applications for this plugin across various tasks. Pertinent examples for a variety of application scenarios are provided in the Section 3.

### 2.2. Validation of the OpenMM Deepmd Plugin

We performed various simulations to validate the Deepmd plugin using different criteria. These tests included conventional MD simulations of water systems within NVE and NVT ensembles. On one hand, we validated the implementation by verifying energy conversation and correcting thermodynamic behavior; on the other hand, we directly compared water properties obtained using this plugin with results from LAMMPS, employing the same DP water model. The extensibility of the plugin was further demonstrated by additional tests, which incorporated the execution of alchemical transformations and adaptive DP/MM simulations. The DP model used for the water system had been trained to investigate the phase diagram of water across a broad range of thermodynamic conditions [44]. This provided a reliable benchmark for assessing the plugin’s accuracy across various phases, which is especially valuable for alchemical simulations that require the DP model’s transferability for water molecules in both coupled and decoupled states. For adaptive DP/MM simulations, we used zinc-containing proteins as examples, employing a DP model trained to correct atomic forces of zinc ions and their coordinated atoms. Simulations on a variety of zinc-containing proteins demonstrated that the zinc coordination structures, such as the tetrahedral coordination structures of the Cys_4_ and the Cys_3_His_1_ groups, can be well reproduced, as detailed in Ref. [62].

#### 2.2.1. Energy Conservation in NVE Simulations

We first validate our implementation by examining energy conservation in microcanonical ensemble (NVE) simulations. Precise energy conservation is crucial in MD simulations, as it ensures that forces are rigorously derived through partial differentiation of the system’s potential energy with respect to atomic coordinates, in accordance with classical Hamiltonian mechanics. In the context of modern deep learning frameworks, this conservation is inherently realized in neural network-based potential energy models, where atomic forces are calculated through automatic differentiation. Nevertheless, in molecular simulations, fluctuations in the total energy throughout the dynamic propagation are induced by the finite time step, denoted as δt. The magnitude of these fluctuations is further influenced by the accuracy of the partial differentiation, the proper implementation of the integrator, and the level of numerical precision exercised within the simulation platform. Consequently, energy conservation serves as an important test for confirming the reliability of simulation results, especially given that practical MD simulations involve sequences of calculations spanning millions, or even billions, of iterations.

For the validation of energy conservation in NVE simulations, an analysis was conducted on a bulk water system containing 256 water molecules within a cubic box measuring 19.80 Å. We note that the water molecule in the DP model is a flexible water model, such that the O-H bonds are not constrained. A Verlet integrator with a 0.2 fs time step was employed throughout the 2 ns NVE simulations using OpenMM and the Deepmd plugin. Both the mixed precision and double precision tests were carried out using the NVIDIA A40 GPU card. As shown in Figure 2, excellent energy conservation profiles were obtained with the OpenMM Deepmd plugin. The magnitudes of fluctuations in the total energy per degree of freedom (DOF) consistently remained significantly below 0.001kcal/mol. Furthermore, no energy drift was observed over the course of the 2 ns simulation trajectories. The standard deviation of the total energy for the entire system was measured to be 0.032kcal/mol for the simulation executed with mixed precision and 0.033kcal/mol for that with double precision. The small difference between mixed and double precision simulations indicates that mixed precision is sufficient for accurately propagating dynamics with MLPs such as the DP model. This is important as mixed precision can provide a significant speedup with negligible loss of precision or energy conservation properties.

#### 2.2.2. Thermodynamic Validation for Canonical Ensemble

The verification of thermodynamic ensemble consistency is equally important, as most MD simulations were carried out with thermostats. Consistency across ensembles under different temperatures provides important validation to ensure that simulations yield reliable results. In 2013, Shirts proposed an elegant technique for assessing thermodynamic ensemble consistency in the context of molecular dynamics or Monte Carlo simulations [66]. This method has since found broad applications, including validating code functionality [67,68] and evaluating the implementation of new force field models [69].

Essentially, provided we perform two equilibrium simulations in the canonical (NVT) ensemble on one system under two different temperatures, $T_{1}$ and $T_{2}$, the potential energy distributions observed within the two simulations should conform to the relationship described by Equation (Equation 2):(2)$$\ln\frac{P\left(E|\beta_{2}\right)}{P\left(E|\beta_{1}\right)}=\left[\beta_{1}-\beta_{2}\right]E+\left[\beta_{2}A_{2}-\beta_{1}A_{1}\right],$$
where $\beta_{1}=\frac{1}{k_{B}T_{1}}$, $\beta_{2}=\frac{1}{k_{B}T_{2}}$, *E* corresponds to the potential energies of the system, and $A_{1}$ and $A_{2}$ represent the Helmholtz free energies at each temperature. Such a linear relationship can be readily tested by extracting potential energy distributions from equilibrated NVT simulation trajectories. One can then fit the logarithm of the quotient against *E* to obtain the slope. Testing the linearity and comparing the computed slope and the theoretical value ($\beta_{1}-\beta_{2}$) provide a rigorous validation on the phase space explored under different temperatures, thereby confirming that the correct thermodynamics ensembles can be obtained.

We performed five independent NVT simulations using this plugin, each running for 8 ns at different temperatures: 300 K, 305 K, 310 K, 315 K, and 320 K, respectively. The Anderson thermostat in OpenMM was employed with a friction coefficient of 1 ps^−1^. The potential energies computed by OpenMM were recorded every 1 ps. From these five NVT trajectories, we generated four sets of potential energy probability ratios corresponding to the temperature pairs 300–305 K, 305–310 K, 310–315 K, and 315–320 K. More specifically, for each temperature pair, we divided the range between the maximum and minimum potential energy values into 30 equal bins. We then calculated the probability for each bin across both trajectories, excluding bins with a probability lower than 0.001.

As shown in Figure 3, the logarithms of energy probability ratios exhibit a very good linear relationship. Least square linear fitting using $\ln\frac{P\left(E|\beta_{2}\right)}{P\left(E|\beta_{1}\right)}$ against *E* resulted in slopes of 0.00663, 0.00606, 0.00627, and 0.00632 mol/kJ, respectively. These values from the simulations compare favorably with the theoretically predicted values ($1/k_{B}T_{1}-1/k_{B}T_{2}$) of 0.00657, 0.00636, 0.00616, and 0.00597 mol/kJ. The relative deviation equals 1% for the 300–305 K pair, while it exhibits a slight increase at higher temperatures.

#### 2.2.3. Structural Property of Bulk Water

While the above two validations, energy conservation and thermodynamic ensemble, primarily focus on the correctness of our model implementation, simulations can also be assessed by comparing the condensed phase properties computed from simulation trajectories with experimental or other computational values. Specifically, we evaluated the structural, kinetic and thermodynamic properties of bulk water using the aforementioned DP water model and the OpenMM Deepmd plugin.

The radial distribution function (RDF) represents a crucial structural property of condensed matter. Theoretically, any thermodynamic properties of condensed matter can be inferred from its RDF, assuming a pairwise additive potential energy function [69,70,71]. RDFs can be obtained via MD simulation trajectories and validated against scattering experimental data. Furthermore, RDFs play a significant role in the parametrization process of empirical force fields for water [72,73,74]. An accurate RDF profile thus demonstrates the capability of the system’s potential energy model to represent the structural properties of the bulk system.

The same cubic box containing 256 water molecules was used for RDF calculations from 4 ns NVT simulations at 298 K, carried out with the OpenMM Deepmd plugin and with LAMMPS. Nosé–Hoover thermostats were used for both OpenMM and LAMMPS simulations for temperature control, with a friction coefficient of 0.5 ps^−1^. The RDFs for oxygen–oxygen, oxygen–hydrogen, and hydrogen–hydrogen pairs were computed using the MDAnalysis package [75] and are shown in Figure 4a–c, respectively. The experimental RDF data were obtained from the work of Soper [76]. The RDF results from the OpenMM simulation and the LAMMPS simulation show remarkable agreement, as evidenced by the nearly identical curves in Figure 4. The first peaks of the RDFs for oxygen–oxygen, oxygen–hydrogen, and hydrogen–hydrogen pairs are positioned at 2.72 Å, 1.75 Å, and 1.56 Å, respectively. These positions align well with the experimental first peaks at 2.73 Å, 1.77 Å, and 1.53 Å. The second peak positions from the simulations are slightly smaller than those in the experimental data, indicating a more compact structure of the bulk water in the simulations.

In addition to conducting NVT simulations with DP models using the plugin, we performed 1.5 ns NPT simulations for a bulk water system consisting of 256 water molecules at 1 bar and 300 K. The pressure was maintained using the Monte Carlo barostat, the temperature was controlled using the Langevin thermostat with a friction coefficient of 1 ps^−1^, and the time step was 0.2 fs. For comparison, we carried out two NPT simulations with the TIP3P water model using the same simulation settings. One simulation employed the rigid TIP3P model, while the other used a flexible model with TIP3P parameters but without constraints. The water density obtained from the NPT simulation with the DP water model was 0.987 ± 0.014 g/mL. In comparison, densities of 0.981 ± 0.018 g/mL and 1.003 ± 0.017 g/mL were obtained with the rigid and flexible TIP3P water models, respectively. We also examined the averaged geometry of water molecules in the bulk system. The O-H distance distributions with the DP water model, for both NVT and NPT simulations, were centered at 0.982 Å with a standard deviation of 0.026 Å. In comparison, the O-H distance is fixed at 0.9572 Å in the rigid TIP3P model, while its distribution is centered at 0.973 Å with a standard deviation of 0.024 Åif a flexible TIP3P model was used in the simulation. A more pronounced difference was observed in the distribution of H-O-H angles. The DP model exhibited H-O-H angles centered at 105.7 ± 5.0 degrees, whereas the flexible TIP3P model showed angles at 102.4 ± 3.2 degrees.

#### 2.2.4. Diffusion Coefficients of Water

Diffusion coefficients represent a crucial kinetic property of bulk liquid systems, offering insights into the behavior of molecules and their interactions with their surroundings. Quantifying the diffusion coefficients thus not only provides essential information on transport behaviors but also serves as a benchmark for assessing the accuracy and reliability of MD simulations [77,78,79]. Similarly, NVT simulations at 298 K using a Nosé–Hoover thermostat were used to compute the diffusion coefficients, with the system size increased to include 4096 water molecules. The mean squared displacements (MSDs) of water molecules were analyzed using the GROMACS analysis tool [80] along MD trajectories. The MSD reflects the mean absolute distance traveled by a molecule as a function of time, and its linear relationship with time is directly proportional to the diffusion coefficient. By fitting the linear relationship, the two trajectories yielded very similar diffusion coefficients: $0.530\times 10^{-5}\;{\mathrm{cm}}^{2}/s$ for LAMMPS and $0.542\times 10^{-5}\;{\mathrm{cm}}^{2}/s$ for OpenMM. While both values are underestimated compared to the experimental data of $2.296\times 10^{-5}\;{\mathrm{cm}}^{2}/s$, their close alignment serves as a validation of our implementation of the DP model in OpenMM.

It is well established that the results of diffusion coefficient calculations based on MD simulations are influenced by finite size effects [81]. To address such an effect, we calculated the diffusion coefficients for bulk water systems comprising 256, 512, 1024, and 2048 water molecules using the OpenMM Deepmd plugin. The resulting diffusion coefficients for these four systems are $0.318\times 10^{-5}\;{\mathrm{cm}}^{2}/s$, $0.459\times 10^{-5}\;{\mathrm{cm}}^{2}/s$, $0.494\times 10^{-5}\;{\mathrm{cm}}^{2}/s$, and $0.500\times 10^{-5}\;{\mathrm{cm}}^{2}/s$, respectively. Following the method of Yeh and Hummer [81], we determined the diffusion coefficient for an infinitely large bulk water system. This was done by fitting the linear relationship between the diffusion coefficient and the inverse of the cubic box size lengths across different bulk water systems. The extrapolated diffusion coefficient for an infinite-sized bulk water system was calculated to be $0.686\times 10^{-5}\;{\mathrm{cm}}^{2}/s$, which is slower than the experimental value of $2.296\times 10^{-5}\;{\mathrm{cm}}^{2}/s$ [82]. In contrast, the widely adopted rigid TIP3P water model tends to overestimate the diffusion coefficient, yielding a computed value of $6.10\times 10^{-5}\;{\mathrm{cm}}^{2}/s$ [81], whereas the popular SPC/E model ($2.432\times 10^{-5}\;{\mathrm{cm}}^{2}/s$) and the flexible SPC/Fw model ($2.359\times 10^{-5}\;{\mathrm{cm}}^{2}/s$) exhibits better agreement with experimental observations [83].

#### 2.2.5. Hydration Free Energy of Water

Going beyond the structural and kinetic properties of bulk water, we further examined a crucial thermodynamic observable, the hydration free energy, to assess the plugin’s functionality. Free energy perturbation calculations, conducted via alchemical transformations, have a broad range of applications in rational drug design and biophysical studies. Therefore, the ability to perform accurate alchemical calculations is essential for validating our implementation.

Following the alchemical simulation protocol discussed in Section 3.2, ten λ states (λ= 0.0, 0.01, 0.03,  0.05, 0.08, 0.12, 0.2, 0.4, 0.7, 1.0) were employed to decouple one water molecule from the others in a bulk water system. For each intermediate λ state, a 4 ns NVT at 300 K simulation was performed with a time step of 0.2 fs, and trajectories were saved every 1 ps. The multistate Bennett acceptance ratio (MBAR) method was used for free energy analysis [84].

Table 1 presents the hydration free energy results from the experiment and the alchemical simulation using the DP model. The uncertainty in $\Delta G_{\mathrm{Hyd}}^{\mathrm{DP}}$ was estimated from five independent replicate alchemical simulations. All replicates converged relatively well over the 4 ns simulation time, as shown in Figure 5a. Sufficient phase space overlap was achieved with 10 λ states, as illustrated by the overlap matrix (Figure 5b) estimated using pymbar [84]. The hydration free energy result from the alchemical simulation using the DP model compares favorably with the experimental value. This also affirms the correctness of the alchemical simulation protocol with the Deepmd plugin, particularly in the application of the addParticlesToDPRegion method.

#### 2.2.6. Performance Profiling

Our implementation is also computationally efficient. Performance profiling of the OpenMM Deepmd plugin was conducted to evaluate its efficiency, alongside a comparison with LAMMPS using the same DP model. This profiling involved simulations of systems with varying particle numbers using both OpenMM and LAMMPS, executed on four CPU cores and one GPU card (Nvidia 1060Ti or A40). As shown in Table 2, the performance of the OpenMM Deepmd plugin is on par with that of LAMMPS across various system sizes. Moreover, the plugin demonstrates scalable performance relative to the number of particles in the system. Additionally, although CPU–GPU communication occurs repeatedly during simulations with the OpenMM Deepmd plugin, the communication time per step is negligible compared to the time taken for evaluating the DP model. To provide a general idea of the performance of the DP model, we note that simulations of a box containing 256 water molecules on a 1060 Ti GPU achieved a rate of 159 ns/day.

## 3. Materials and Methods

The DeepPotentialModel class, through the versatile methods it contains, affords substantial flexibility for deploying the DP model within OpenMM across a variety of application scenarios. In this section, we detail three distinct examples that demonstrate the utility of the deep potential model in OpenMM via this plugin. These examples include conventional MD simulations, alchemical simulations related to free energy calculations, and hybrid MLP/MM simulations that are often employed for biomolecular systems.

### 3.1. Conventional MD Simulations with the DP Model

The most common application of the deep potential model within OpenMM is to carry out MD simulations. The DeepPotentialModel class provides a streamlined approach for setting up the simulation system that exclusively uses the DP model. The setup can be completed with four lines of code in Listing 1.

**Listing 1.** Conventional MD simulations with the DP model.

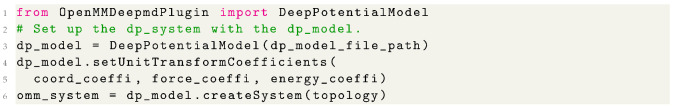



Further customizations to the simulation system can be applied to the omm_system object. MD simulations are generally employed to explore equilibrated properties, and all OpenMM thermostats and barostats are supported for simulations in either the canonical ensemble or the isothermal–isobaric ensemble.

### 3.2. Alchemical Simulations with the DP Model

To quantify the free energy difference between two equilibrium states, alchemical simulation techniques, which interpolate between two Hamiltonians, are often employed. These alchemical simulations typically entail stepwise transitions between two states to estimate the differences in their free energies, as Figure 6 presents. In classical force field-based simulations, the smooth interpolation of force field parameters can facilitate such transitions. However, such interpolation schemes are not directly transferable to simulations that use DP models. The neural network parameters optimized within a DP model are not explicitly interpretable in physical terms. Attempting to transition these parameters could lead to discontinuities in the forces and energies predicted by the model. As a result, alchemical simulations with DP models require a direct transition of the total potential energies, as highlighted in previous works [86]. This is equivalent to progressively turning off the non-bonded interaction as a whole between two parts of the system, instead of turning off the electrostatic and van der Waals interactions sequentially. Although this approach sacrifices some computational efficiency, it enables accurate and reliable estimations of free energy differences when using DP models.

The potential energies of the system transition between two states during the alchemical simulation are defined as in Equation (Equation 3).
(3)$$U=\lambda \cdot U_{A}+\left(1-\lambda \right)\cdot U_{B}=\lambda \cdot U_{A}+\left(1-\lambda \right)\left(U_{B_{1}}\left(\left\{j\right\}\right)+U_{B_{2}}\left(\left\{i\right\}\right)\right),$$
where $U_{A}$ and $U_{B}$ denote the potential energy of the system in the two states, and λ serves to regulate the system’s intermediate states. If we consider $U_{A}$ and $U_{B}$ as corresponding to the coupled and decoupled states for two groups of particles, respectively, then the potential energy for the decoupled states can be further expressed as the sum of $U_{B_{1}}$ and $U_{B_{2}}$, which encapsulate the potential energies for the two decoupled groups of particles, *i* and *j*.

With OpenMM, alchemical simulations utilizing DP models can be conducted via a plugin method that conveniently partitions the system into two distinct particle groups and imposes a scaling factor λ to the forces and energy as predicted by the DP model for each group. This approach enables dual-topology simulations under various thermodynamic conditions, which allows DP-based free energy calculations.

To formulate the decoupled state in this example, the addParticlesToDPRegion function was used to divide the system into two particle groups, as shown in Listing 2. These groups were then individually treated by two separate DP models. Moreover, combining the addParticlesToDPRegion function with traditional molecular mechanics force fields available in OpenMM facilitates the implementation of hybrid DP/MM simulations.

**Listing 2.** Alchemical simulations with the DP model.

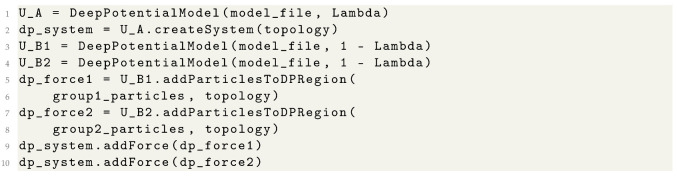



### 3.3. Hybrid DP/MM Simulations with Fixed or Adaptive DP Regions

MLPs are increasingly employed in a range of applications for material systems [87,88,89,90], yet their applications to biomolecular systems remain limited due to the absence of long-range interactions in MLPs. Consequently, the hybrid MLP/MM scheme emerges as a more practical approach for modeling biomolecules with MLPs. In this scheme, the MLP is utilized to delineate the intramolecular interactions [65,91,92] or accuracy-critical reactions [48,63,93] for a subset of the entire simulation system.

To perform a DP/MM simulation using this plugin, users can employ the addParticlesToDPRegion function to specify the input particles for the DP model (see Listing 3). This function allows the selected particles to serve both as the coordinate input for the DP model and as the recipients of predicted DP forces throughout the simulation.

**Listing 3.** DP/MM simulations with pre-selected particle.

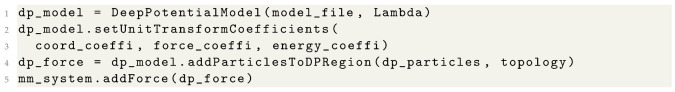



Beyond facilitating DP/MM simulations with predefined particles, this plugin also supports adaptive DP/MM simulations. Here, particles for the DP region are automatically selected based on a predefined radius and a set of explicitly defined central particles [62]. In this adaptive scheme, the DP region extends the particle set to include those within the radius of the central particles, as well as particles belonging to the same residue of selected ones. Thus, the central particles and selection radius are required when employing adaptive DP/MM simulations, see Listing 4. This allows for crucial interactions to be accounted for within biomolecular systems.

**Listing 4.** DP/MM simulations with adaptive selected particles.

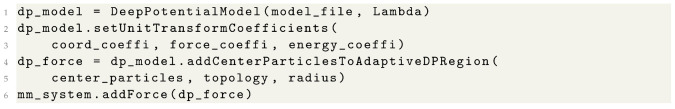



Different selection methods within the hybrid DP/MM model cater to diverse application needs. For example, the DP model with pre-selected particles can be deployed either independently [65,91,92,93] or alongside QM/MM methods [48,63,64] to depict the intramolecular interactions for the subset of interests. When used in isolation, interactions among these particles modeled by classical force fields can be disregarded prior to simulation. However, in adaptive DP/MM simulations, ignoring such intramolecular interactions is less straightforward, necessitating the combined use of the DP model and classical force fields [62]. We note that the energy conservation and the sampling efficiency associated with adaptive DP/MM schemes remain open issues and are subject to further investigations.

These examples showcase the plugin’s flexibility and user-friendliness for setting up and performing simulations that integrate the DP model with classical force fields in OpenMM. Moreover, the plugin’s compatibility with other advanced simulation techniques like QM/MM opens up new avenues for applications.

## 4. Discussion

The integration of MLPs with popular MD engines is increasingly important given the rapid advancement of MLPs in various systems. However, the flexible design of MLP architectures and frameworks means that there is no one-size-fits-all approach for their integration with simulation software. Furthermore, with the community proposing new application scenarios for MLPs, especially hybrid MLP/MM models [48,61,62,63,64,65,94], an efficient and adaptable approach for embedding MLPs alongside other force fields within MD engines becomes essential. Here, the integration of MLPs with MD software is well illustrated by the implementation and validation of the OpenMM Deepmd plugin. This plugin has been made openly available at https://github.com/JingHuangLab/openmm_deepmd_plugin, along with test examples (accessed on 13 Januray 2024). It facilitates not only conventional MD simulations using the deep potential model but also supports hybrid models that incorporate only part of the simulation system into the DP model. The design of the OpenMM Deepmd plugin allows for a flexible combination of DP with other FF models, a feature vital for the effective application of MLPs in biomolecular systems [62,65].

## 5. Conclusions

The OpenMM Deepmd plugin integrates the increasingly popular DP model into the well-established MD engine, OpenMM. To enhance the DP model’s applicability within the OpenMM framework, a user-friendly class named DeepPotentialModel has been implemented. This class enables the construction of various application scenarios, including conventional, alchemical, and hybrid simulations with either fixed or adaptively selected regions. A range of tests was conducted to validate the plugin’s implementation. Regarding simulation rigor, we examined both energy conservation in NVE simulations and thermodynamic ensemble consistency in NVT simulations. Additionally, several bulk water properties were verified, representing the structural (RDFs), kinetic (diffusion coefficient), and thermodynamic (hydration free energy) characteristics of the simulation system. We showed that the properties computed using the OpenMM Deepmd plugin and the LAMMPS software align well, and both compare favorably with experimental values.

The reliability of the deep potential model’s application in OpenMM forms the basis for the various tests conducted on this plugin. To extend the applicability of the DP model to a wide range of scenarios within the OpenMM framework, a user-friendly class named DeepPotentialModel has been implemented. We are confident that this plugin will prove to be a valuable asset for both the OpenMM and the DP communities, facilitating the expanded use of machine learning potential models in molecular and biomolecular simulations.

## Figures and Tables

**Figure 1 ijms-25-01448-f001:**
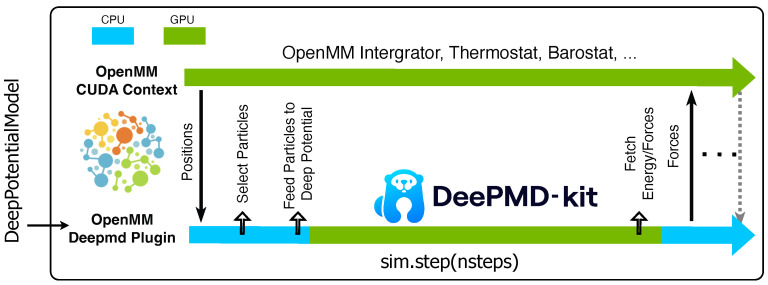
The architecture of the OpenMM Deepmd plugin, illustrating the data flow and the interaction between OpenMM and the DP model.

**Figure 2 ijms-25-01448-f002:**
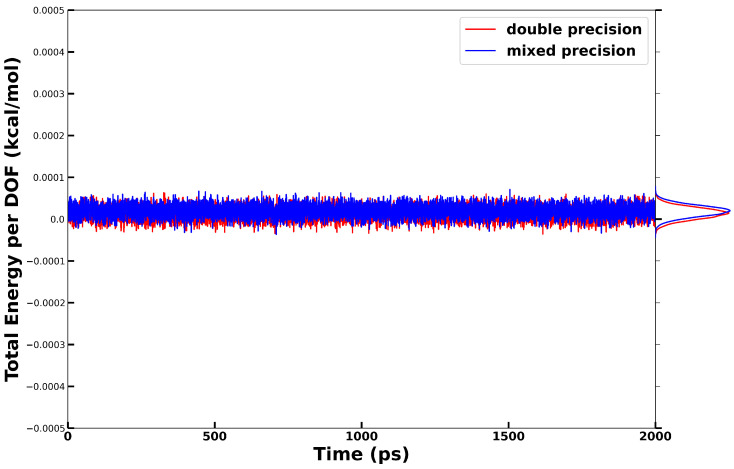
Energy conservation during NVE simulations using the OpenMM Deepmd plugin.

**Figure 3 ijms-25-01448-f003:**
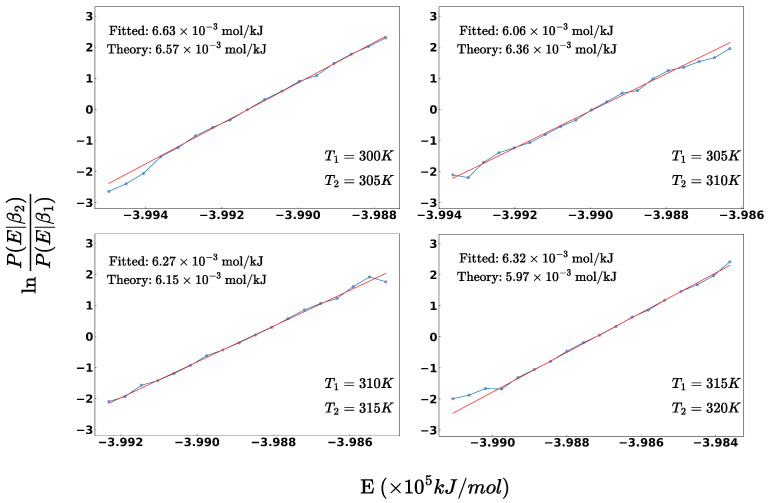
The ratio of the potential energy distributions sampled from NVT simulations at two different temperatures, represented by blue dots. Four temperature pairs were examined, each demonstrating a strong linear relationship. The slopes from linear fitting (red) are compared with their corresponding theoretical values. To ensure a more reliable estimation of the slope for the linear relationship of the energy distribution, the first and last three bins were excluded from the fitting process.

**Figure 4 ijms-25-01448-f004:**
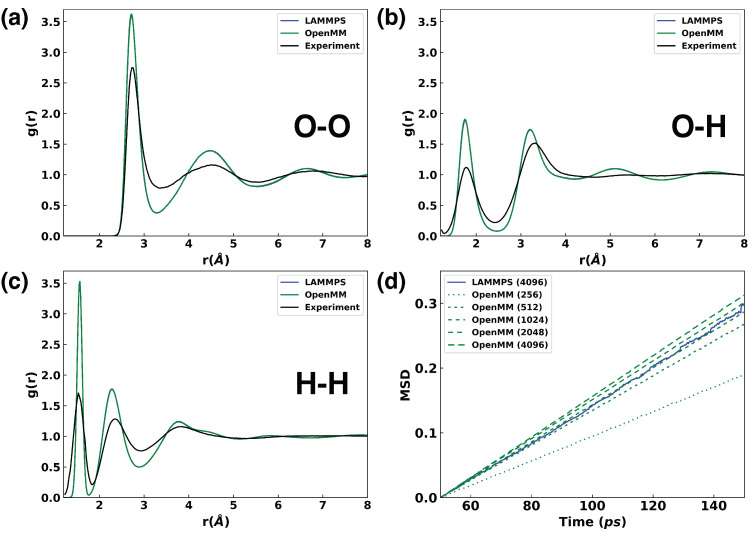
The structural properties (RDFs) and kinetic properties (MSDs) of bulk water were obtained using the OpenMM Deepmd plugin and LAMMPS. (**a**) water O-O RDFs; (**b**) water O-H RDFs; (**c**) water H-H RDFs; (**d**) water MSDs. Both the OpenMM Deepmd plugin and LAMMPS employed the same DP model in the simulations.

**Figure 5 ijms-25-01448-f005:**
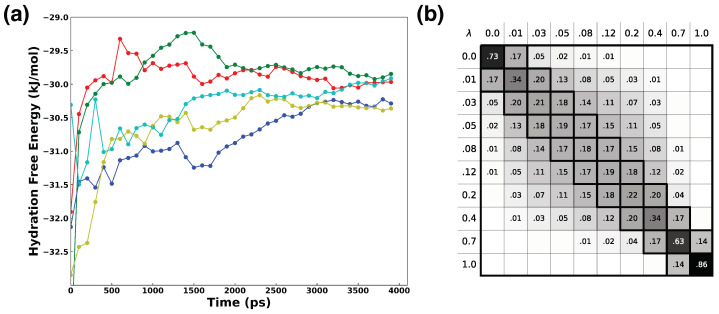
(**a**) Convergence of free energy estimates over the lengths of five independent replicates of alchemical simulations using the DP water model. (**b**) Overlap matrix O between the 10 λ states for the 1st replicate alchemical simulation, estimated using pymbar. Each element of the matrix $O_{ij}$ represents an estimate of the probability of observing a sample from state *i* in state *j*.

**Figure 6 ijms-25-01448-f006:**
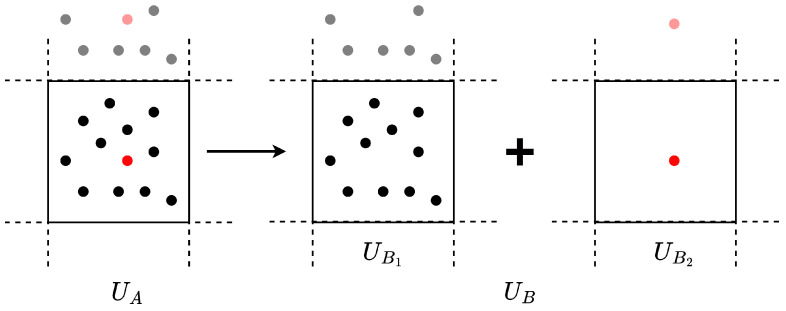
Alchemical transition from the coupled state $U_{A}$ to the decoupled state $U_{B}$.

**Table 1 ijms-25-01448-t001:** Hydration free energy of water. $\Delta G_{\mathrm{Hyd}}^{\mathrm{DP}}$ represents the average from 5 replicate simulations, with the value in parentheses indicating its uncertainty. $\Delta G_{\mathrm{Hyd}}^{\mathrm{TIP}3P*}$ was calculated using the same alchemical protocol described in Section 3.2, without employing the softcore potential, to validate this protocol. $\Delta G_{\mathrm{Hyd}}^{\mathrm{TIP}3P}$ and $\Delta G_{\mathrm{Hyd}}^{\mathrm{Exp}}$ values are sourced from ref [85]. Units in kJ/mol.

$\Delta G_{\mathrm{Hyd}}^{\mathrm{Exp}}$	$\Delta G_{\mathrm{Hyd}}^{\mathrm{DP}}$	$\Delta G_{\mathrm{Hyd}}^{\mathrm{TIP}3P*}$	$\Delta G_{\mathrm{Hyd}}^{\mathrm{TIP}3P}$
−26.5	−30.08(0.23)	−24.33	−25.3

**Table 2 ijms-25-01448-t002:** Simulation performance of the OpenMM Deepmd plugin with various numbers of particles.

Hardware	Num. of Particles	OpenMM (ns/day)	DP Evaluation Cost (μs)	Communication Cost (μs)	LAMMPS (ns/day)
4 CPU cores	192	1.84	7682.650	113.673	1.895
+ 1060 Ti	768	0.554	31,201.579	121.686	0.569
	3072	0.139	123,432.237	149.248	0.149
	6144	0.0687	250,992.324	171.257	0.076
4 CPU cores	192	2.73	6884.679	360.378	2.924
+ A40	768	1.51	11,500.541	361.795	1.376
	3072	0.482	35,651.468	291.818	0.420
	6144	0.243	69,671.011	328.957	0.228
	9216	0.159	105,681.057	345.710	0.159
	12,288	0.121	143,443.777	370.841	0.123
	15,360	0.093	180,575.597	400.271	0.100
	18,432	0.074	228,041.861	426.523	0.084
	21,504	0.060	286,185.872	453.118	0.073
	24,576	0.056	312,219.517	488.567	0.063

## Data Availability

The source code and data presented in this study are openly available at https://github.com/JingHuangLab/openmm_deepmd_plugin. All simulation scripts used in this manuscript can be found in the sub-directory python/tests (accessed 13 January 2024).

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
