# Peer review of "Implementation and Validation of an OpenMM Plugin for the Deep Potential Representation of Potential Energy"

_ijms, 2024, doi:10.3390/ijms25031448_

Round 1
Reviewer 1 Report
Comments and Suggestions for Authors
Here, I provide my review of the manuscript "Implementation and Validation of an OpenMM Plugin for the Deep Potential Representation of Potential Energy”. Therein, as the name states, the authors present an implementation of the deep potential (DP) model for the OpenMM simulation framework. The manuscript is generally interesting and well-written. Nevertheless, I suggest some rewriting, to clarify the structure of the document. Furthermore, I suggest some additional tests of the implementation, as described below.
I organised my comments in enumerated bullet points:
- The authors decided against a conservative structure of a classical research manuscript. Instead of introduction, methods, results and so on, they chose different sections. I generally understand this decision, as they present an implementation rather than an deductive research paper, based on novel data. However, in the current form they often jump between description of an analyses method, showing and discussing results related to this method, and then moving on to the next analysis method. I would suggest restructuring in terms of a section that includes all the descriptions of the methodology and another section with all the related results.
On the same point: I think the performance benchmark results (table 2) are misplaced in the discussion and conclusion section. To me, they belong to the same section as all the other results. - I noticed that the authors never simulated in NPT. I think it would be a valuable add-on to investigate the density of water in NPT and compare to regular MD.
- In Figure 5, the authors compare LAMMPS and OpenMM with the same computational model, right? First, I thought that the authors compare some regular MD water model with DP water. Maybe this comparison may be clarified a little bit.
- For the comparison of the diff. coefficient, did the authors use a flexible water model, or did they use regular rigid TIP3P? Because the DP water is flexible, I think it would be fair to compare with a flexible water model.
- I would also be interested in the averaged geometry of the DP water: distribution of OH bond distances and HOH-bond angles. Perhaps, the authors want to add this to the supplementary information.
- Even though, this might not be a flattering comparison, I think the performance benchmarks should also include regular MD water simulations, e.g., with (flexible) TIP3P. The readers will want to know how expensive the ML potential is in comparison to the regular MD potentials.
Minor points:
- Considering Figure 6: Personally, I would not call this “time evolution”. This term suggests that we look at a time-dependent property that varies at different points in time. However, the authors actually want to show the convergence of the estimated value with increasing simulation time that is evaluated.
- The y-scale in Figure 3 is not chosen optimally.
- Since the authors are publishing an implementation here, I would personally suggest to make the availability of the code and some test examples more prominent. I would mention the GitHub repository and that there are some tests available there.
Reviewer 2 Report
Comments and Suggestions for Authors
The article "Implementation and Validation of an OpenMM Plugin for the Deep Potential Representation of Potential Energy" is devoted to practical application of deep learning in application to molecular simulations.
The authors consider the applicability of the method and the developed plugin using the example of short trajectories of water molecules. Unfortunately, it is not clear whether the method can be used on more complex systems: proteins, membranes, DNA, etc.? In my opinion, the article can be published after major revision.
Equations 1
What in R?
Not only does the DP model provide an accurate description of the energy landscape for a 70
wide range of chemical space, but also enables simulations of systems that are otherwise 71
inaccessible by ab initio calculations [43]. 72
Like any method, the method described in this article has a certain range of applicability and disadvantages. Please add them to the article.
2.3.2. Alchemical Simulations with the DP model 221
Alchemical calculations involve the sequential removal of electrostatic and van der Waals interactions. It is not clear how the lambda states change for your model?
Such a linear relation- 372
ship can be readily tested by extracting potential energy distributions from equilibrated 373
NVT simulation trajectories. One can then fit the logarithm of the quotient against E to 374
How exactly were the potential energy distributions extracted? Please add a description so that other researchers can reproduce the simulations.
The labels in Figure 4 are too small and unreadable.
The slopes derived from the linear fitting of simulation data are 0.00663, 0.00606, 390
0.00627, and 0.00632 mol/kJ, respectively. They compare favorably with the theoretically 391
predicted values of 0.00657, 0.00636, 0.00616, and 0.00597 mol/kJ. The relative deviation 392
It is not clear how the data for the linear fitting and the theoretically predicted values are calculated.
linear relationship with time is directly proportional to the diffusion coefficient. By fit- 435
ting the linear relationship, the two trajectories yielded very similar diffusion coefficients: 436
0.530 [1] 10ô€€€5cm2/s for LAMMPS and 0.542 [1] 10ô€€€5cm2/s for OpenMM. 437
Please add comparison with experimental value here. The values are greatly underestimated compared to experimental data. Therefore, the coincidence of the two calculated values gives a very imaginary advantage to the method.
The extrapolated diffusion 447
coefficient for an infinite-sized bulk water system was calculated to be 0.686 [1] 10ô€€€5cm2/s, 448
which is slower than the experimental value of 2.296 [1] 10ô€€€5cm2/s [91]. In contrast, the 449
commonly used TIP3P water model overestimates the diffusion coefficient with a computed 450
value of 6.10 [1] 10ô€€€5cm2/s [90].
The attempt to explain the underestimation of the value by comparison with another model (TIP3P water) is not clear. For example, for SPC/E water at a given temperature, excellent agreement with experiment is obtained using classical MD with AMBER FF. Perhaps it is worth adding a comparison with this model of water to your work?
Following the alchemical simulation protocol discussed in Section 2.3.2, ten l states 460
(l = 0.0, 0.01, 0.03, 0.05, 0.08, 0.12, 0.2, 0.4, 0.7, 1.0) were employed to decouple one 461
In [93] for the LJ part of the free energy, 15λ values were used; for the electrostatic component five λ points were used. Did using just 10 lambda points provide you with enough area overlap? Provide the appropriate graphs.
Reviewer 3 Report
Comments and Suggestions for Authors
This manuscript introduces an OpenMM plugin for the Deep Potential model which is useful for molecular dynamics simulations with the Deep Potential machine learning model or hybrid ML/MM model. This is a welcome addition to the literature and potentially publishable in IJMS. I only have a few minor comments.
(1) All figures and listings should be referenced in the main text to improve readability.
(2) Unit conversion factors are used to convert DP units to OpenMM units. In the code snippets, the authors used variables such as "coord_coeffi". I'm wondering whether there are predefined variables for these coefficients in the plugin? I think that would make the plugin more user friendly. OpenMM also has some predefined coefficients, e.g. OpenMM::NmPerAngstrom http://docs.openmm.org/latest/userguide/theory/01_introduction.html
(3) Line 253-254: It's not clear to me why dp_model1 and dp_model2 both use 1-Lambda. Please provide more details about this example.
(4) Line 296-297: "can be deployed either independentlyindependently".
(5) As the authors pointed out, one main application of this plugin is for hybrid ML/MM simulations.So it would be helpful to have some validation for hybrid simulations. Especially in the case of adaptive ML region, how to ensure energy conservation?
Round 2
Reviewer 1 Report
Comments and Suggestions for Authors
I think this is a nice contribution. Thank you.
I am accepting this manuscript for publication. Nevertheless, I still think that the y-scale in Figure 3 is not well chosen. I leave it to the authors to decide if and how to change that eventually.
Reviewer 2 Report
Comments and Suggestions for Authors
The majority the issues have been addressed. I recommend the revised version of the manuscript for publication in the journal.